# Protocol of a prospective comprehensive evaluation of an elastic band beard cover for filtering facepiece respirators in healthcare

**Daryl Lindsay Williams**[1,2], **Benjamin Kave**[1], **Charles Bodas**[3], **Fiona Begg**[3], **Megan Roberts**[3], **Irene Ng**[1,2]*

1 Department of Anaesthesia and Pain Management, Royal Melbourne Hospital, Parkville, Australia,
2 Melbourne Medical School, University of Melbourne, Parkville, Victoria, Australia, 3 Respiratory Protection Program, Royal Melbourne Hospital, Parkville, Australia

* Irene.Ng@mh.org.au

**Data Availability Statement:** No datasets were generated or analysed in this study protocol. All the individual de-identified data that support the findings of this proposed study will be available

## Abstract

Individuals who are unable to be clean shaven for religious, medical or cultural reasons are unable to wear a filtering facepiece respirator (FFR), as the respirator cannot provide adequate protection against aerosol-transmissible diseases. There is currently a paucity of validated techniques to ensure the safe inclusion of bearded healthcare workers in the pandemic workforce. We propose to undertake a healthcare-based multi-modal evaluation study on the elastic band beard cover for FFR technique, examining the quantitative fit test (QNFT) results, usability and skill level of participants with repeated assessments over time. This is a prospective study conducted through the Respiratory Protection Program at the Royal Melbourne Hospital. Healthcare workers are invited to participate if they require respiratory protection and cannot shave for religious, cultural or medical reasons. An online education package on the use of respiratory protective equipment and the elastic band beard cover for FFR technique is provided. This is followed by a face-to-face session, where the participant will receive: one-on-one training; undergo a skill assessment on their donning, doffing and user seal check techniques; complete QNFTs and a usability survey. Participants will be invited to repeat the assessment within 3 months of the first session and at 12 months. This study involves multimodal and repeated assessments of an elastic band beard cover for FFRs. The findings of this study will provide information on: whether this simple technique can provide safe, consistent and effective respiratory protection; whether it will interfere with occupational activities; and whether it is comfortable and tolerable for the duration of wear. This is of significant importance to the health workforce around the world, who cannot shave but require access to respiratory protective equipment during the COVID-19 pandemic.

## Introduction

Airborne contaminants can range from several micrometres down to fractions of a micrometre, whereas human hair has an average thickness of about 100 microns [1]. Facial hair of

upon request from the corresponding author. Data will be available immediately following publication of the result findings until five years after publication.

**Funding:** This study is supported by the Australian Victorian State Government, who provides funding for the implementation of the Respiratory Protection Program, and ongoing education, training, evaluation and research in relation to respiratory protection equipment. The funder has no role in study design, data collection and analysis, decision to publish, or preparation of the manuscript.

**Competing interests:** The authors have declared that no competing interests exist.

greater than 1mm in length is known to decrease respiratory protection when using tight fitting respirators due to disruption of the face seal [2]. International standards agencies recommend that individuals who have stubble, a moustache, sideburns, or a beard, which passes between the skin and the sealing surface should not wear a tight fitting respirator that requires a facial seal, whether full or half facepiece [3–5]. Individuals who are unable to be clean shaven for religious, medical or cultural reasons are unable to wear a filtering facepiece respirator (FFR), as the respirator cannot provide adequate protection against aerosol-transmissible diseases. The requirement of staff to be clean shaven to effectively wear a FFR may be considered indirect discrimination against those who are unable to shave for cultural or religious reasons, even though it is likely to be a genuine and lawful request by employers [6].

An alternative form of respiratory protective equipment in unshaven workers is to wear a loose-fitting powered air purifying respirator (PAPR). However, PAPRs have several issues related to the need for extensive user training, equipment maintenance and storage, cleaning and disinfection processes, lack of source control, battery life, noise impact and visual field interference [7]. The use of PAPRs therefore need careful consideration and extensive consultation with relevant stakeholders before being implemented [7].

Recently, an under-mask elastic band beard cover, also known as the Singh Thattha technique [8], has been identified as a potential option for those who are unable to shave. This technique involves the use of a long elastic band placed over the beard and stretched tightly around the sides of the face, and tied on top of the head. The tight-fitting respirator is then worn with the seal formed on the band (artificial skin) rather than across facial hair. The original study by Singh et al [8], included a small cohort of participants, looking at both qualitative fit testing (QLFT) and quantitative fit testing (QNFT). The QLFT cohort included 27 participants, recording a 92.6% (25/27) pass rate using predominantly reusable elastomeric respirators. However, in healthcare settings, the commonest respirator used is disposable FFR. Only three participants were assessed with FFRs in Singh's study [8] and two failed the QLFT. Only five bearded participants were assessed using QNFT achieving a 100% pass rate, but none of the participants used disposable FFRs.

Health authorities and occupational health organisations do not currently recommend the adoption of the Singh Thattha technique, due to "limited evidence in published studies that provides assurance that a good seal to the face is achieved and maintained or that the seal and protection is predictable, reliable and that the wearer is therefore adequately protected" [9].

To address this research gap, we propose to undertake a study on the elastic band beard cover technique, using a multi-modality evaluation process on healthcare workers, based on the framework recommended by the Centers for Disease Control and Prevention (CDC) and the National Institute for Occupational Safety and Health (NIOSH) in project BREATHE for respiratory protective equipment [10]. This includes quantitative fit testing, skill assessment, user assessment, and repeat assessments. The findings of this study will provide information on whether the combination of FFR and elastic band can provide safe and effective respiratory protection, whether it will interfere with occupational activities, whether it is comfortable and tolerable for the duration of wear, and the impact of facial hair growth over time on QNFT results whilst using the technique.

## Materials and methods

This is a prospective study conducted through the Respiratory Protection Program at the Royal Melbourne Hospital. This study was approved by Melbourne Health Human Research Ethics Committee (QA2022022) and the individual in this manuscript has given written informed consent (as outlined in PLOS consent form) to publish his photograph. Healthcare

workers and healthcare worker students are invited to participate in this study if they require respiratory protection at work from airborne biohazards at P2/N95 level; cannot shave for religious, cultural or medical reasons; and are able to wear an elastic band wrap around the head for the purpose of achieving a face seal with a tight-fitting N95/P2 respirator at work. We exclude healthcare workers who choose not to be clean shaven in the face-seal zone of their respirators due to reasons other than religious, cultural or medical. Use of tight-fitting respirators with facial hair is against international standards [4,11] and manufacturer's instructions [12]. Requiring employees to shave for respiratory protection should be considered a reasonable request under Occupational Health and Safety laws [13,14]. The use of elastic band beard cover technique should be prioritised for healthcare workers who cannot shave due to tenable reasons, such as religious, cultural or medical restrictions.

Eligible Australian healthcare workers will receive an invitation email that outlines the objectives and requirement of the study. Participation is voluntary. Potential internal participants are identified by completing an unable to shave attestation and approval is obtained from their manager. External participants are referred by the worker's human resource or occupational health and safety director at their respective organisation.

Interested participants complete an online survey, hosted by REDCap 10.5.2 (Vanderbilt University, Nashville, Tennessee, USA), via a link on their invitation email. The survey includes basic demographic information, work hazard assessment, health safety screen, training and experience with N95/P2 FFR, and attitude assessment (Appendix 1 in S1 File). Consent is implied upon survey completion. We then provide an online education package, which includes written material and video presentations on respiratory protective equipment and the elastic band technique.

Participants then attend a face-to-face session in a controlled non-clinical environment, where they:

1. Receive one-on-one respirator training;

2. Have their beard length measured;

3. Undergo a baseline QNFT without the elastic band on;

4. Receive one-on-one training on the use of the elastic band beard cover technique;

5. Undergo three consecutive QNFTs with the elastic band on, for two types of N95/P2 FFR (repeat donning and doffing procedure for each QNFT);

6. Undergo skill assessment on their donning, doffing and user seal check techniques;

7. Complete usability assessment.

A standard operating procedure will be used for the elastic band beard cover technique, to ensure that a consistent technique is used and assessed for the study. The principles behind the standard operating procedure development are shown in Appendix 2 in S2 File. The technique involves the use of an elastic band wrap (TheraBand® professional resistance band, Thera-Band, Akron, OH, USA), which is 10–15 cm wide and cut to one-metre in length (Fig 1). A green TheraBand that expands to 100% of its length at 2.1 kg of force [15] will be used to achieve a smooth surface whilst allowing free movement of the mandible during speech. Prior to use, the TheraBand is cleaned with disinfectant wipes or a damp cloth. The width of head covering if present, is to be minimised as much as practicable to avoid overstretching the straps of the FFR and to allow the TheraBand to sit flush on the face. In the Sikh population, all Turbans will be limited to 3 metres in length or alternatively an under-Turban head covering will be used. The TheraBand will cover the facial hair over the chin and cheeks, cross the ears and

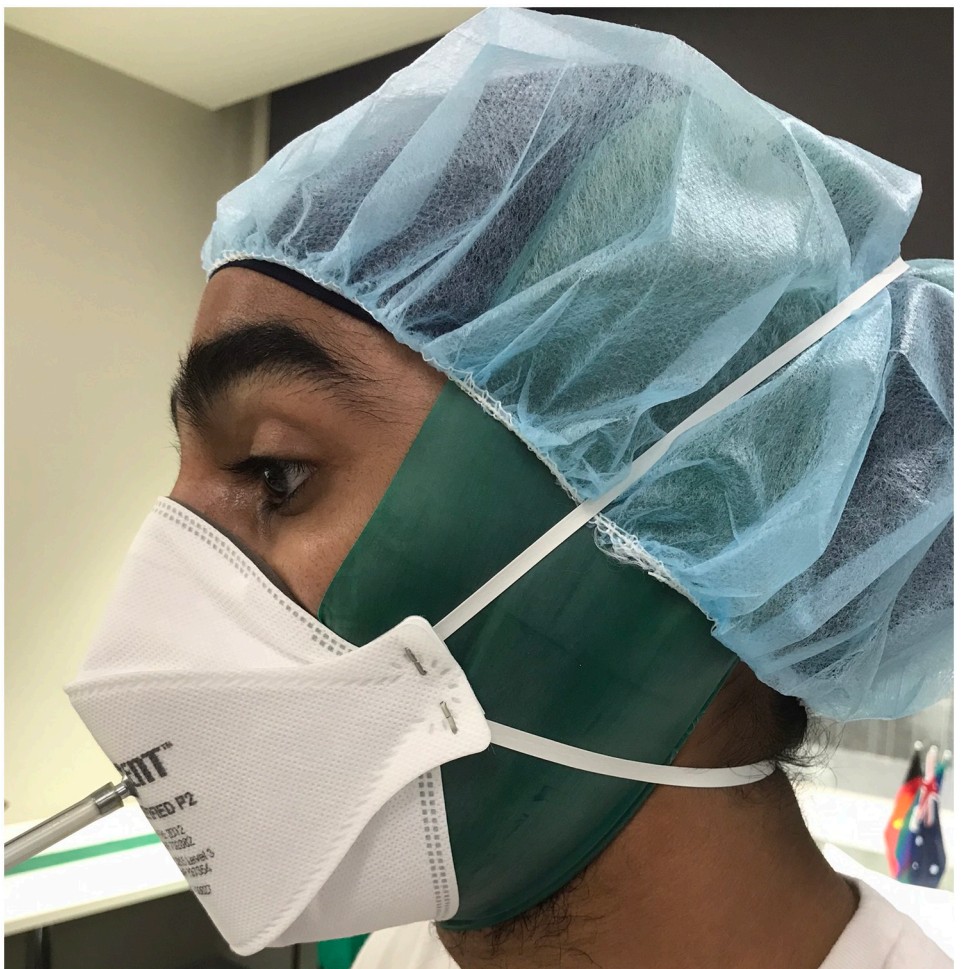

**Fig 1. Elastic band beard cover.** A person wearing elastic band beard cover while undergoing quantitative fit test for P2 respirator.

be tied 3cm (+/- 2cm) in front of the crown of the head. The band must be tight enough to sit flat on the chin and face, so that the TheraBand merges with the skin of the participant in the face-seal area. A disposable head cover is applied over the TheraBand before a FFR is donned, to cover the knot of elastic band so as to minimise the risk of entanglement with the head straps of the respirator, and also to minimise potential contamination of the Turban or under-Turban (if present).

We will use two makes of three-panel flat-fold N95/P2 FFRs for this study, i.e. 3M 1870 + Aura (3M, St. Paul, MN, USA) and Trident™ P2 respirator (Industree, West Gosford, NSW, Australia). These two types of FFRs are chosen because as of 4th March 2022, they are the most readily available N95/P2 FFRs in our jurisdiction [16], and they have been shown to have very high QNFT pass rates and high comfort and usability ratings in healthcare workers [17,18]. The participant will perform donning, user seal check and doffing of the N95/P2 FFR as per manufacturers' instruction.

The doffing sequence follows the usual infection prevention practices, with the aim to minimise complexity and cognitive load, whilst being able to be performed without the assistance of a second person. The N95/P2 respirator is doffed as per manufacturer guidelines and hand

hygiene is performed. The disposable head cover is discarded and hand hygiene is performed. The TheraBand is removed by gently sliding the head strap component anteriorly until the elastic band loosens from under the chin, with care not to touch the face components of the elastic band. The knot on the crown of the head is kept tied during the doffing. The TheraBand is discarded in an appropriate clinical waste bin. Hand hygiene is then performed.

The QNFT is performed in our dedicated Respiratory Protection Program fit testing facility with controlled room temperature and strict infection control procedures. It involves the use of the Portacount machine (PortaCount® Pro+ 8048, TSI Incorporated, St Paul, Minnesota, USA), which is calibrated at the beginning of each testing day, using a provided HEPA filter. The subject will attach their mask probe to the PortaCount machine via plastic tubing that has been previously soaked in 99.5% isopropyl alcohol solution and allowed to dry. The test will then begin according to the United States Occupational Safety and Health Administration protocols [19], which consists of four exercises:

1. Bending over at the hips and returning to upright repeatedly, taking two breaths when bent over. This lasts 50 seconds;

2. Reading a standardised text aloud for 30 seconds;

3. Moving the head from side to side for 30 seconds;

4. Flexing and extending the neck for 30 seconds.

A baseline QNFT assessment without the TheraBand (naked beard) will be completed first, followed by three consecutive QFNTs with the TheraBand on, for each make of FFR. The participant is required to repeat donning and doffing procedure (including the TheraBand) for each of the three QNFTs. The test will be observed throughout by personnel trained in the use of the PortaCount machine and any breach of protocol will be addressed by recommencing the test. The respirator fit is determined by the fit factor, which is calculated by dividing the concentration of the particles in ambient air outside the mask by that inside the mask. An overall fit factor of >100 is considered a pass. The overall fit factor is calculated as the harmonic mean of the fit factor achieved for each individual exercise.

Each participant will have their donning, doffing and user seal check techniques assessed by direct observation by a trained fit tester, using a standardised marking sheet (Appendix 3 in S3 File). A similar marking system has been used in our previous study, which showed good consistency [20]. Inter- and intra-rater reliability will be assessed in this study, using a series of video recordings of ten separate donning and doffing procedures with pre-determined embedded errors.

At the conclusion of the face-to-face session, the participants will be required to complete a usability assessment via a survey link on their mobile device (Appendix 4 in S4 File). The development and validation of a similar tool is described in our previous study [21].

The multimodal assessment process will be repeated within 3 months of initial assessment and at 12 months. This includes repeating the three consecutive QNFTs wearing the elastic band beard cover; having their donning, doffing and user seal check techniques assessed by a qualified fit tester; and also completing the usability assessment.

Our primary aim is to examine whether the elastic band beard cover technique can reliably provide adequate respiratory protection. Therefore, the primary outcome of this study is to investigate the percentage of participants, with the TheraBand on, who can achieve three consecutive QNFT passes (overall fit factor > 100 for each of the three tests) with either of the two types of FFR tested. Secondary outcomes include the percentage of participants who can achieve a first-time QNFT pass with the TheraBand on; comparison of first-time QNFT pass

rates and fit factors (overall and individual fit factors) between the elastic band beard cover technique and the naked beard; skill assessment results; and usability survey results. We will also investigate whether there is any association between beard length and overall fit factor. We will repeat the assessment at 3 months and 12 months, including the percentage of participants who can achieve three consecutive QNFT passes with the TheraBand on, skill assessment and usability assessment results.

## Statistical analysis

Descriptive statistics will be used to present the demographic data, QNFT results, skill assessment results and usability assessment results. McNemar's test will be used to compare the QNFT pass rates, and Wilcoxon signed ranks test will be used to compare the fit factors between TheraBand elastic band beard cover technique and the naked beard. Pearson correlation will be used to investigate the association between beard length and overall fit factor. Kappa statistics will be used to assess inter- and intra-rater reliability in the skill assessment. The kappa value is interpreted to indicate consistency, as follows: <0.21 was "poor", 0.21 to 0.40 was "fair", 0.41–0.60 was "moderate", 0.61–0.8 was "good" and 0.81–1.0 was "very good" [22]. Statistical analysis will be performed using Stata 13.0 (Statacorp, College Station, Texas, USA).

There is no sample size calculation for this evaluation study. We will attempt to acquire the largest possible convenience sample. It is dependent on the number of suitable candidates from various Victorian healthcare organisations. However, a previous study showed that no full-bearded healthcare workers achieved a FFR fit [23]. If we conservatively assume a 10% baseline QNFT pass rate for our participants using the 3M Aura 1870+ without the elastic band beard cover and that it improves to 50% with the elastic band cover, then we would need about 25 participants for a power of 0.8.

## Discussion

Tight-fitting respirator face masks, such as FFRs, are considered the reference standard respiratory protective equipment for healthcare workers who may be exposed to aerosol transmitted diseases [3–6,24]. The research protocol proposed in this paper would deliver a relatively large-scale multimodal and repeated assessments of an elastic band beard cover for FFRs. This is of significant importance to the health workforce, as it provides a path forward for the many healthcare workers around the world who elect not to shave and have had difficulty negotiating the suitability of respiratory protective equipment during the COVID-19 pandemic [8,25].

The strength of this protocol is that it uses the principles and consensus standards of Project BREATHE [10] to validate the elastic band technique. Better Respiratory Equipment Using Advanced Technologies for Healthcare Employees (Project BREATHE), established in 2009, is an inter-agency working group of the United States Federal Government, whose purpose is to develop a set of consensus recommendations to improve the respiratory protective equipment used by healthcare workers [10]. The Project's 28 recommendations provide a framework for the development of respiratory protective equipment, ensuring that: 1) Respirators should perform their intended functions safely and effectively; 2) Respirators should support, not interfere with workplace activities; 3) Respirators should be comfortable and tolerable for the duration of the wear; and 4) Respiratory protection programs should comply with government standards and local policies.

Our protocol's evaluation of the elastic band beard cover technique aligns with the above standards, by undertaking a multimodal assessment, which measures safety, effectiveness,

usability and skill level of participants with repeated assessments over time. Our approach is standardised, thereby minimising variation and maximising safety for healthcare workers.

Improved workforce participation is an expected benefit of this protocol. In some studies, up to 46% of the male health workforce had facial hair capable of interfering with respirator fit, and in those with a full beard, none were able to achieve a pass on QNFT [23]. In addition to this, recognition of the problem amongst healthcare workers is poor, with many bearded subjects not being aware that their safety was jeopardised [23,26]. Validation of the elastic band technique will empower this cohort of workers to maximise their respiratory protection and participate fully in the workforce. As a consequence, previous guidelines that recommended redeployment of bearded employees to low-risk settings will hopefully not be necessary.

A further consideration in adopting this protocol is the ability to avoid any implication of discrimination against bearded healthcare workers, who may wear a beard for religious, ethnic or cultural reasons. Published legal opinions have determined that an employer likely can require employees to be clean shaven for their health and safety, but it is preferable to avoid any appearance of discrimination by considering alternatives to redeployment [25].

The practice of respiratory protection in hospitals has undergone significant evolution over the two years of the COVID-19 pandemic. As we move into the future, a respiratory protection solution for bearded men has the capacity to improve staff satisfaction and workplace safety without compromising workplace participation.

We acknowledge that there are limitations to our protocol. Firstly, in an effort to standardise our approach, only one elastic band is being utilised. This may exclude other viable alternatives and lead to supply constraints in the event of a surge in demand. However, we believe that standardising the approach to the extent possible is necessary in the first serious attempt to validate this method.

Secondly, we are only utilising two types of FFRs, both of which have demonstrated high fit test pass rates in other studies [17,18]. If we can successfully validate the technique using these two types of FFR, we would recommend further research with a broader range of N95/P2 masks. Thirdly, our cohort size may be not be powered to determine smaller effects of the elastic band on mask fit in different components of the QNFT procedure, but our aim in this study is to investigate the viability and efficacy of the technique.

## Conclusions

In conclusion, there is currently a paucity of validated techniques to ensure the safe inclusion of bearded healthcare workers in the pandemic workforce. There are considerable benefits in pursuing the validation of a simple and cheap solution, namely the elastic band beard cover technique, and our protocol aims to achieve this to the highest published standards available.

## Supporting information

**S1 File. Appendix 1.** Baseline survey.
(PDF)

**S2 File. Appendix 2.** Principles of Standard of Operation.
(PDF)

**S3 File. Appendix 3.** Skill assessment marking sheet.
(PDF)

**S4 File. Appendix 4.** Usability assessment survey.
(PDF)

**S5 File.**
(PDF)

## Acknowledgments

We would like to thank all the staff from the Royal Melbourne Hospital Respiratory Protection Program for their support of this project.

## Author Contributions

**Conceptualization:** Daryl Lindsay Williams, Benjamin Kave, Charles Bodas, Fiona Begg, Megan Roberts, Irene Ng.

**Data curation:** Daryl Lindsay Williams, Charles Bodas, Fiona Begg, Megan Roberts, Irene Ng.

**Formal analysis:** Megan Roberts, Irene Ng.

**Funding acquisition:** Daryl Lindsay Williams, Charles Bodas, Fiona Begg.

**Investigation:** Daryl Lindsay Williams, Benjamin Kave, Charles Bodas, Fiona Begg, Megan Roberts.

**Methodology:** Daryl Lindsay Williams, Benjamin Kave, Charles Bodas, Fiona Begg, Megan Roberts, Irene Ng.

**Project administration:** Daryl Lindsay Williams, Charles Bodas, Fiona Begg.

**Resources:** Daryl Lindsay Williams, Benjamin Kave, Charles Bodas, Fiona Begg.

**Software:** Charles Bodas, Fiona Begg.

**Supervision:** Daryl Lindsay Williams, Charles Bodas, Fiona Begg, Megan Roberts.

**Validation:** Daryl Lindsay Williams, Benjamin Kave, Charles Bodas, Fiona Begg, Megan Roberts, Irene Ng.

**Visualization:** Irene Ng.

**Writing – original draft:** Daryl Lindsay Williams, Benjamin Kave, Irene Ng.

**Writing – review & editing:** Daryl Lindsay Williams, Benjamin Kave, Charles Bodas, Fiona Begg, Megan Roberts, Irene Ng.

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
