## [Decision Letter · Decision Letter 0]

26 Sep 2022

PONE-D-22-13621Protocol of a prospective comprehensive evaluation of an elastic band beard cover for filtering facepiece respirators in healthcarePLOS ONE

Dear Dr. Ng,

Thank you for submitting your manuscript to PLOS ONE. After careful consideration, we feel that it has merit but does not fully meet PLOS ONE’s publication criteria as it currently stands. Therefore, we invite you to submit a revised version of the manuscript that addresses the points raised during the review process.

The manuscript has been evaluated by three reviewers, and their comments are available below.

The reviewers have raised a number of concerns that need attention. They request additional information on the data availability and clarity on how this study contributes to the existing literature. 

Could you please revise the manuscript to carefully address the concerns raised?

We look forward to receiving your revised manuscript.

Kind regards,

Alice Coles-Aldridge

Editorial Office

PLOS ONE

Journal Requirements:

3. We note that Figure 1 includes an image of a participant in the study. 

Reviewers' comments:

Reviewer's Responses to Questions

**Comments to the Author**

1. Does the manuscript provide a valid rationale for the proposed study, with clearly identified and justified research questions?

Reviewer #1: Yes

Reviewer #2: Yes

Reviewer #3: Yes

2. Is the protocol technically sound and planned in a manner that will lead to a meaningful outcome and allow testing the stated hypotheses?

Reviewer #1: Yes

Reviewer #2: Yes

Reviewer #3: Partly

3. Is the methodology feasible and described in sufficient detail to allow the work to be replicable?

Reviewer #1: Yes

Reviewer #2: Yes

Reviewer #3: Yes

4. Have the authors described where all data underlying the findings will be made available when the study is complete?

Reviewer #1: Yes

Reviewer #2: No

Reviewer #3: Yes

5. Is the manuscript presented in an intelligible fashion and written in standard English?

Reviewer #1: Yes

Reviewer #2: Yes

Reviewer #3: Yes

6. Review Comments to the Author

You may also provide optional suggestions and comments to authors that they might find helpful in planning their study.

Reviewer #1: The authors have presented a protocol for a prospective multimodal evaluation of an elastic beard cover for filtering facepiece respirators (FFR) involving quantitative fit testing at 0,3 and 12 months. The protocol is well considered and presented clearly with appropriate justification, namely the lack of comprehensive evaluation of the suitability of elastic beard covers in healthcare settings. Relevant literature is cited.

I have only a few minor comments;

1. Methods p7. Is there methodological justification for the exclusion of healthcare workers who are not clean shaven for reasons other than religious, cultural or medical? This should be clarified.

2. Methods p6-. The authors may wish to consider grading or quantifying beard lengths unless they only intend to recruit participants with ‘full beards’, which should be explicitly stated if it is the case. Degree of facial hair has been correlated with FFR fit and may impact the study results.

Reviewer #2: Thank you for the opportunity to review this protocol. I found it well written, well-reasoned, and that the authors were forthcoming about any foreseeable limitations and how these are mitigated (e.g., the likely small sample sizes). The surveys were well constructed and, assuming sufficient sample sizes, I think they will address the stated study aims. I also think the focus of the proposed project is timely and important given the increased interest in bioaerosols and PPE efficacy that arose due to SARS-COV2.

My only minor critique has to do with the lack of information about data sharing. Data sharing/availability is key part of every PLOS review, and I found the statement “All relevant data from this study will be made available upon study completion” to be rather vague. Shared how? Via appendices? Via dryad? Will the data be shared publicly and freely without restriction? What does “relevant” mean? Given the choice to submit this protocol to PLOS, I think the authors need to add additional clarity on the scope and manner of their intended data sharing in the body of the protocol. Beyond this I have no further revisions to recommend.

Reviewer #3: Although the study is technically sound, given the recent publication by Bhatia et al. (see Findings and citation below), my concern is that the research gap described on page 6 is no longer the primary motivation for the study. The abstract discussion also indicates “This is the first large-scale multimodal assessment in the literature of an elastic band beard cover for FFRs” which may no longer be accurate.

Given the potential for success with the elastic band technique and the models of respirator proposed to be tested (see data from table 2 below), the study might be characterized as a replication of an existing study, with a longer-term evaluation of efficacy.

One idea for a novel intervention would be to test the use of a balaclava rather than an elastic exercise band (https://www.amazon.com/dp/B01LE02EHQ/). As someone who has worn the elastic band for testing and study purposes, user comfort is an issue to consider. The elastic band is not a garment intended to be worn and captures sweat with extended wear. It also can create discomfort based on positioning under the chin and how tightly it is knotted atop the head. On the other hand, a balaclava is intended to be worn for cold weather or adventure activities and may tolerate sweat and extended wear more favorably. It is a garment that can be worn in multiple styles (over the head or not, over the nose, under the chin) and so could be adjusted to cover the beard but not the nose and mouth of the user. Because it does not require tying or other modes of securing to the head, it may represent an easier to use intervention. Testing for whether this intervention works with disposable respirators and collecting information on the likelihood of usage, comfort, tolerability, etc. (either vs. an exercise band or alone) would represent a novel study proposal. Again, replication is important for science, but extending research and testing new ideas is also important.

Findings: Thirty subjects were assessed; of these, 24 (80%) passed quantitative fit testing with at least one tight-fitting P2/N95 disposable respirator. Among these subjects, the median best-achieved fit factor was 200 (interquartile range 178-200). None of the subjects had an adverse reaction to the under-mask beard cover.

3M 1860 14/28 (50) 170 (128–200)

3M 1870+ 7/20 (35) 199 (181–200)

Trident RTCFFP2 15/22 (68) 200 (191–200)

Bhatia DDS, Bhatia KS, Saluja T, Saluja APS, Thind A, Bamra A, Singh G, Singh N, Clezy K, Dempsey K, Hudson B, Jain S. Under-mask beard covers achieve an adequate seal with tight-fitting disposable respirators using quantitative fit testing. J Hosp Infect. 2022 May 31;128:8-12. doi: 10.1016/j.jhin.2022.05.015. Epub ahead of print. PMID: 35662553.

7. PLOS authors have the option to publish the peer review history of their article (what does this mean?). If published, this will include your full peer review and any attached files.

Reviewer #1: No

Reviewer #2: No

Reviewer #3: **Yes: **Steven E. Prince

---

## [Author Response · Author response to Decision Letter 0]

3 Oct 2022

Response to Reviewers

Journal Requirements:

Response: Apologies. We have now revised the manuscript so that it meets PLOS ONE’s style requirements.

Response: We have moved the ethics statement to the Methods section. Thank you for the suggestion.

3. We note that Figure 1 includes an image of a participant in the study. 

Response: We have obtained a signed consent from the individual whose image is shown in Figure 1. We have also included a statement to clarify this in the method section.

Reviewers' comments:

Reviewer's Responses to Questions

Comments to the Author

1. Does the manuscript provide a valid rationale for the proposed study, with clearly identified and justified research questions?

Reviewer #1: Yes

Reviewer #2: Yes

Reviewer #3: Yes

2. Is the protocol technically sound and planned in a manner that will lead to a meaningful outcome and allow testing the stated hypotheses?

Reviewer #1: Yes

Reviewer #2: Yes

Reviewer #3: Partly

3. Is the methodology feasible and described in sufficient detail to allow the work to be replicable?

Reviewer #1: Yes

Reviewer #2: Yes

Reviewer #3: Yes

4. Have the authors described where all data underlying the findings will be made available when the study is complete?

Reviewer #1: Yes

Reviewer #2: No

Reviewer #3: Yes

5. Is the manuscript presented in an intelligible fashion and written in standard English?

Reviewer #1: Yes

Reviewer #2: Yes

Reviewer #3: Yes

6. Review Comments to the Author

You may also provide optional suggestions and comments to authors that they might find helpful in planning their study.

Reviewer #1: The authors have presented a protocol for a prospective multimodal evaluation of an elastic beard cover for filtering facepiece respirators (FFR) involving quantitative fit testing at 0,3 and 12 months. The protocol is well considered and presented clearly with appropriate justification, namely the lack of comprehensive evaluation of the suitability of elastic beard covers in healthcare settings. Relevant literature is cited.

I have only a few minor comments;

1. Methods p7. Is there methodological justification for the exclusion of healthcare workers who are not clean shaven for reasons other than religious, cultural or medical? This should be clarified.

Response: Use of tight-fitting respirators with facial hair is against international standards (OSHA Respiratory Protection Standard, 29 CFR 1910.134)1 and manufacturer’s instructions2. As shown in Prince et al.’s study3, facial hair under a P2/N95 respirator can diminish respiratory protection and potentially jeopardise healthcare workers’ and patient safety. We believe that requiring employees to shave for respiratory protection is a reasonable request under OHS law4,5. We should show compassion to healthcare workers who may experience a loss of identity or skin irritation with daily shaving. However, the use of elastic band beard cover technique should be prioritised for healthcare workers who cannot shave with tenable reasons, such as religious, cultural or medical restrictions, which are the inclusion criteria for this study. This helps ensure that this group of healthcare workers are not discriminated against and potentially alleviate pressure from workforce shortage. 

We have added the justification for the exclusion criteria in the methods section of the protocol.

1 https://www.cdc.gov/niosh/npptl/pdfs/facialhairwmask11282017-508.pdf

2 https://multimedia.3m.com/mws/media/1682579O/facial-hair-and-respirator-fit-testing-policy-technical-bulletin.pdf

3 Prince SE, Chen H, Tong H, et al. Assessing the effect of beard hair lengths on face masks used as personal protective equipment during the COVID-19 pandemic. Journal of exposure science & environmental epidemiology. 2021:1-8.

4 https://www.worksafe.vic.gov.au/occupational-health-and-safety-your-legal-duties

5 https://www.legislation.vic.gov.au/in-force/acts/occupational-health-and-safety-act-2004/042

2. Methods p6-. The authors may wish to consider grading or quantifying beard lengths unless they only intend to recruit participants with ‘full beards’, which should be explicitly stated if it is the case. Degree of facial hair has been correlated with FFR fit and may impact the study results.

Response: Thank you for the comment. Yes, we are measuring beard length each time the participant is having a quantitative fit test. Apologies for not mentioning this in the original protocol. We also believe that the density of the hair follicles and the grooming style may also have an impact on respirator fit. However, we are not collecting these data in this study. We anticipate that almost all our participants would be fully bearded because they would have grown their beard for a while with religious, cultural or medical restrictions.

We have now added beard length measurement as one of the data outcomes in the methods and statistical analysis sections of the protocol.

Reviewer #2: Thank you for the opportunity to review this protocol. I found it well written, well-reasoned, and that the authors were forthcoming about any foreseeable limitations and how these are mitigated (e.g., the likely small sample sizes). The surveys were well constructed and, assuming sufficient sample sizes, I think they will address the stated study aims. I also think the focus of the proposed project is timely and important given the increased interest in bioaerosols and PPE efficacy that arose due to SARS-COV2.

My only minor critique has to do with the lack of information about data sharing. Data sharing/availability is key part of every PLOS review, and I found the statement “All relevant data from this study will be made available upon study completion” to be rather vague. Shared how? Via appendices? Via dryad? Will the data be shared publicly and freely without restriction? What does “relevant” mean? Given the choice to submit this protocol to PLOS, I think the authors need to add additional clarity on the scope and manner of their intended data sharing in the body of the protocol. Beyond this I have no further revisions to recommend.

Response: Thank you for the positive comments. Apologies for the vague data sharing statement. We have now revised the statement, which says - “All the individual de-identified data that support the findings of this proposed study will be available upon request from the corresponding author. Data will be available immediately following publication of the result findings until five years after publication. We have included this statement in the protocol. 

Reviewer #3: Although the study is technically sound, given the recent publication by Bhatia et al. (see Findings and citation below), my concern is that the research gap described on page 6 is no longer the primary motivation for the study. The abstract discussion also indicates “This is the first large-scale multimodal assessment in the literature of an elastic band beard cover for FFRs” which may no longer be accurate.

Given the potential for success with the elastic band technique and the models of respirator proposed to be tested (see data from table 2 below), the study might be characterized as a replication of an existing study, with a longer-term evaluation of efficacy.

Response: Thank you for the comments. We agree that our study proposal is no longer the first in the literature to assess quantitative fit test results of the elastic band beard cover technique with a reasonable sample size. Our study is different from Bhatia’s study in several ways. Firstly, unlike Bhatia’s study, our methodology is strictly standardized, including donning/doffing technique, training process, types of respirators used and sequence of the fit testing procedure. Moreover, we have other components in the evaluation, including skill assessment, usability assessment and repeated assessments over time. We anticipate that our sample size will be larger than Bhatia’s study as well. Therefore, we believe that the findings of our study will provide additional and valuable information in the literature.

We have now revised both the abstract and the main text discussion sections to ensure that we do not state our study is the first to perform fit test assessment of the elastic band beard cover technique.

One idea for a novel intervention would be to test the use of a balaclava rather than an elastic exercise band (https://www.amazon.com/dp/B01LE02EHQ/). As someone who has worn the elastic band for testing and study purposes, user comfort is an issue to consider. The elastic band is not a garment intended to be worn and captures sweat with extended wear. It also can create discomfort based on positioning under the chin and how tightly it is knotted atop the head. On the other hand, a balaclava is intended to be worn for cold weather or adventure activities and may tolerate sweat and extended wear more favorably. It is a garment that can be worn in multiple styles (over the head or not, over the nose, under the chin) and so could be adjusted to cover the beard but not the nose and mouth of the user. Because it does not require tying or other modes of securing to the head, it may represent an easier to use intervention. Testing for whether this intervention works with disposable respirators and collecting information on the likelihood of usage, comfort, tolerability, etc. (either vs. an exercise band or alone) would represent a novel study proposal. Again, replication is important for science, but extending research and testing new ideas is also important.

Response: Thank you for the suggestions. We agree that evaluating fit test results with the use of balaclava would be a novel intervention and that wearing a balaclava may be more comfortable than wearing an elastic band. We believe that elastic band is much more effective than balaclava because elastic band has an impermeable membrane, therefore acting like an artificial skin while providing an occlusive seal. In our protocol the elastic band is discarded after each use. On the other hand, a balaclava is permeable and can fold or create creases easily. Bhatia did measure the quantitative fit test pass rate with balaclava and found that the pass rate was only 17/30, 57%. (Ref: Appendix 4A in Clinical Excellence Commission. Respiratory Protection Program Manual. July 2022. Version 1.1. https://www.cec.health.nsw.gov.au/__data/assets/pdf_file/0004/696712/Respiratory-Protection-Program-Manual.pdf). We are hoping for QNFT pass rates to be greater than 80% using our standardised elastic band protocol and therefore, we think that the use of balaclava is probably not a technique that is broadly applicable for healthcare workers in clinical environment. 

We have considered usability and safety while designing this study. We ensure that the participant will apply a disposable head cover over the elastic band before a respirator is donned, to cover the knot of elastic band so as to minimise the risk of entanglement with the head straps of the respirator, and also to minimise potential contamination of both the elastic band and the Turban or under-Turban (if present).

We acknowledge that elastic band may cause discomfort with extended use due to heat and moisture build up. We therefore included comfort and tolerability assessments in this study. Moreover, we are currently designing a separate study to measure heat and moisture levels with the use of elastic band under a respirator. We hope to gather more scientific information in this area.

Findings: Thirty subjects were assessed; of these, 24 (80%) passed quantitative fit testing with at least one tight-fitting P2/N95 disposable respirator. Among these subjects, the median best-achieved fit factor was 200 (interquartile range 178-200). None of the subjects had an adverse reaction to the under-mask beard cover.

3M 1860 14/28 (50) 170 (128–200)

3M 1870+ 7/20 (35) 199 (181–200)

Trident RTCFFP2 15/22 (68) 200 (191–200)

Bhatia DDS, Bhatia KS, Saluja T, Saluja APS, Thind A, Bamra A, Singh G, Singh N, Clezy K, Dempsey K, Hudson B, Jain S. Under-mask beard covers achieve an adequate seal with tight-fitting disposable respirators using quantitative fit testing. J Hosp Infect. 2022 May 31;128:8-12. doi: 10.1016/j.jhin.2022.05.015. Epub ahead of print. PMID: 35662553.

7. PLOS authors have the option to publish the peer review history of their article (what does this mean?). If published, this will include your full peer review and any attached files.

Do you want your identity to be public for this peer review? For information about this choice, including consent withdrawal, please see our Privacy Policy.

Reviewer #1: No

Reviewer #2: No

Reviewer #3: Yes: Steven E. Prince

---

## [Decision Letter · Decision Letter 1]

16 Jan 2023

Protocol of a prospective comprehensive evaluation of an elastic band beard cover for filtering facepiece respirators in healthcare

PONE-D-22-13621R1

Dear Dr. Ng,

We’re pleased to inform you that your manuscript has been judged scientifically suitable for publication and will be formally accepted for publication once it meets all outstanding technical requirements.

Kind regards,

Binson V A, Ph.D.

Guest Editor

PLOS ONE

Additional Editor Comments (optional):

Reviewers' comments:

Reviewer's Responses to Questions

**Comments to the Author**

1. Does the manuscript provide a valid rationale for the proposed study, with clearly identified and justified research questions?

Reviewer #1: Yes

Reviewer #3: Yes

2. Is the protocol technically sound and planned in a manner that will lead to a meaningful outcome and allow testing the stated hypotheses?

Reviewer #1: Yes

Reviewer #3: Yes

3. Is the methodology feasible and described in sufficient detail to allow the work to be replicable?

Reviewer #1: Yes

Reviewer #3: Yes

4. Have the authors described where all data underlying the findings will be made available when the study is complete?

Reviewer #1: Yes

Reviewer #3: Yes

5. Is the manuscript presented in an intelligible fashion and written in standard English?

Reviewer #1: Yes

Reviewer #3: Yes

6. Review Comments to the Author

You may also provide optional suggestions and comments to authors that they might find helpful in planning their study.

Reviewer #1: The authors have responded satisfactorily to the reviewer comments in my opinion and I have no further comments. I look forward to the subsequent publication of their study results.

Reviewer #3: The balaclava used in the document referenced by the authors is a disposable "medical" permeable version that indeed looks subject to folding or creasing. It is not a garment per se, but more closely related to an elastic hairnet.

In place of this disposable option, my suggestion was to use a tight fitting reusable fabric balaclava garment, such as worn for cold weather sports and designed to fit tightly around the face.

There are many variations and fabrics available for this type of garment. I am by no means suggesting there is a "correct" fabric balaclava. Rather, that the form factor may provide for easier donning (sliding a tightly fitting reusable garment over the head and positioning to cover the beard) versus wrapping and tying the elastic band.

I fully agree that the goal is to create the best artificial skin in order to better restore the seal with the mask. In fact, when considering alternatives to the elastic bands tested in our study, we briefly researched artificial skin used by tattoo artists for practice purposes, ultimately deciding that it was not a practicable solution.

Back to the protocol, I am not sure why the elastic band needs to be discarded after use. The CEC manual suggests disinfection after use is an option, allowing for reuse. But perhaps it is preferred for infection control or hygiene purposes in this particular study.

Regardless of these details, the study is important and will provide practical information for respirator users. I think the authors have done a solid job of creating a testing protocol and experimental design to answer an important question. Given that they are considering the moisture and temperature impacts of the elastic band solution in a separate study, they are aware of potential barriers to adoption. In trying to predict whether ease of use and user comfort are factors that could facilitate adoption for an off the shelf garment that does not need to be tied and is designed to be worn for multiple hours during sports related activities. It is also quite possible that such a garment is uncomfortable in indoor settings and temperatures. But it is worth considering in my opinion, in case it has the potential to remove a barrier for usage. Although the present study focuses on healthcare workers there are many bearded males, particularly in the younger demographic, who might benefit from knowledge of a practical intervention for improving respirator fit. Again, this is my opinion that data would generalize to society more broadly with an easier to implement intervention. That being said, an elastic band solution could be made easier to use with Velcro or an alternative method to secure to the top of the head, which could remove a barrier to more widespread adoption.

Response: Thank you for the suggestions. We agree that evaluating fit test results with

the use of balaclava would be a novel intervention and that wearing a balaclava may

be more comfortable than wearing an elastic band. We believe that elastic band is

much more effective than balaclava because elastic band has an impermeable

membrane, therefore acting like an artificial skin while providing an occlusive seal. In

our protocol the elastic band is discarded after each use. On the other hand, a

balaclava is permeable and can fold or create creases easily. Bhatia did measure the

quantitative fit test pass rate with balaclava and found that the pass rate was only

17/30, 57%. (Ref: Appendix 4A in Clinical Excellence Commission. Respiratory

Protection Program Manual. July 2022. Version 1.1.

https://www.cec.health.nsw.gov.au/__data/assets/pdf_file/0004/696712/Respiratory-

Protection-Program-Manual.pdf). We are hoping for QNFT pass rates to be greater

than 80% using our standardised elastic band protocol and therefore, we think that the

use of balaclava is probably not a technique that is broadly applicable for healthcare

workers in clinical environment.

7. PLOS authors have the option to publish the peer review history of their article (what does this mean?). If published, this will include your full peer review and any attached files.

Reviewer #1: No

Reviewer #3: **Yes: **Steven E. Prince

---

## [Editor Report · Acceptance letter]

23 Jan 2023

PONE-D-22-13621R1 

Protocol of a prospective comprehensive
 evaluation of an elastic band beard cover for filtering facepiece respirators in healthcare 

Dear Dr. Ng:

I'm pleased to inform you that your manuscript has been deemed suitable for publication in PLOS ONE. Congratulations! Your manuscript is now with our production department. 

Kind regards, 

on behalf of

Dr. Binson V A 

Guest Editor

PLOS ONE